# DISCRIMINATIVE k-SHOT LEARNING USING PROBABILISTIC MODELS

## ABSTRACT

This paper introduces a probabilistic framework for k-shot image classification. The goal is to generalise from an initial large-scale classification task to a separate task comprising new classes and small numbers of examples. The new approach not only leverages the feature-based representation learned by a neural network from the initial task (representational transfer), but also information about the classes (concept transfer). The concept information is encapsulated in a probabilistic model for the final layer weights of the neural network which acts as a prior for probabilistic k-shot learning. We show that even a simple probabilistic model achieves state-of-the-art on a standard k-shot learning dataset by a large margin. Moreover, it is able to accurately model uncertainty, leading to well calibrated classifiers, and is easily extensible and flexible, unlike many recent approaches to k-shot learning.

## 1 INTRODUCTION

A child encountering images of helicopters for the first time is able to generalize to instances with radically different appearance from only a handful of labelled examples. This remarkable feat is supported in part by a high-level feature-representation of images acquired from past experience. However, it is likely that information about previously learned concepts, such as aeroplanes and vehicles, is also leveraged (e.g. that sets of features like tails and rotors or objects like pilots/drivers are likely to appear in new images). The goal of this paper is to build machine systems for performing k-shot learning, which leverage both existing feature representations of the inputs and existing class information that have both been honed by learning from large amounts of labelled data.

K-shot learning has enjoyed a recent resurgence in the academic community (Lake et al., 2015; Koch et al., 2015; Vinyals et al., 2016; Snell et al., 2017; Srivastava & Salakhutdinov, 2013). Current state-of-the-art methods use complex deep learning architectures and claim that learning good features for k-shot learning entails *training for k-shot specifically* via episodic training that simulates many k-shot tasks. In contrast, this paper proposes a general framework based upon the combination of a deep feature extractor, trained on batch classification, and traditional probabilistic modelling. It subsumes two existing approaches in this vein (Srivastava & Salakhutdinov, 2013; Burgess et al., 2016), and is motivated by similar ideas from multi-task learning (Bakker & Heskes, 2003). The intuition is that deep learning will learn powerful *feature representations*, whereas probabilistic inference will transfer top-down *conceptual information* from old classes. Representational learning is driven by the large number of training examples from the original classes making it amenable to standard deep learning. In contrast, the transfer of conceptual information to the new classes relies on a relatively small number of existing classes and k-shot data points, which means probabilistic inference is appropriate.

While generalisation accuracy is often the key objective when training a classifier, calibration is also a fundamental concern in many applications such as decision making for autonomous driving and medicine. Here, calibration refers to the agreement between a classifier's uncertainty and the frequency of its mistakes, which has recently received increased attention. For example, Guo et al., 2017 show that the calibration of deep architectures deteriorates as depth and complexity increase. Calibration is closely related to catastrophic forgetting in continual learning. However, to our knowledge, uncertainty has so far been over-looked by the k-shot community even though it is high in this setting.

Our basic setup mimics that of the motivating example above: a standard deep convolutional neural network (CNN) is trained on a large labelled training set. This learns a rich representation of images at the top hidden layer of the CNN. Accumulated knowledge about classes is embodied in the top

layer softmax weights of the network. This information is extracted by training a probabilistic model on these weights. K-shot learning can then 1) use the representation of images provided by the CNN as input to a new softmax function, 2) learn the new softmax weights by combining prior information about their likely form derived from the original dataset with the k-shot likelihood.

**The main contributions of our paper are:**

1) We propose a probabilistic framework for k-shot learning. It combines deep convolutional features with a probabilistic model that treats the top-level weights of a neural network as data, which can be used to regularize the weights at k-shot time in a principled Bayesian fashion. We show that the framework recovers $L_2$-regularised logistic regression, with an automatically determined setting of the regularisation parameter, as a special case.

2) We show that our approach achieves state-of-the-art results on the *mini*ImageNet dataset by a wide margin of roughly $6\%$ for 1- and 5-shot learning. We further show that architectures with better batch classification accuracy also provide features which generalize better at k-shot time. This finding is contrary to the current belief that episodic training is necessary for good performance and puts the success of recent complex deep learning approaches to k-shot learning into context.

3) We show on *mini*ImageNet and CIFAR-100 that our framework achieves a good trade-off between classification accuracy and calibration, and it strikes a good balance between learning new classes and forgetting the old ones.

## 2 PROBABILISTIC K-SHOT LEARNING

**K-shot learning task.** We consider the following discriminative k-shot learning task: First, we receive a large dataset $\widetilde{\mathcal{D}} = \{\widetilde{\mathbf{u}}_i, \widetilde{y}_i\}_{i=1}^{\widetilde{N}}$ of images $\widetilde{\mathbf{u}}_i$ and labels $\widetilde{y}_i \in \{1, \ldots, \widetilde{C}\}$ that indicate which of the $\widetilde{C}$ classes each image belongs to. Second, we receive a small dataset $\mathcal{D} = \{\mathbf{u}_i, y_i\}_{i=1}^{N}$ of $C$ new classes, $y_i \in \{\widetilde{C} + 1, \widetilde{C} + C\}$, with $k$ images from each new class. Our goal is to construct a model that can leverage the information in $\widetilde{\mathcal{D}}$ and $\mathcal{D}$ to predict well on unseen images $\mathbf{u}^*$ from the new classes; the performance is evaluated against ground truth labels $y^*$.

**Summary.** In contrast to several recent k-shot learning approaches that mimic the k-shot learning task by episodic training on simulated k-shot tasks, we propose to use the large dataset $\widetilde{\mathcal{D}}$ to train a powerful feature extractor on batch classification, which can then be used in conjunction with a simple probabilistic model to perform k-shot learning. In 2003, Bakker & Heskes introduced a general probabilistic framework for multi-task learning with multi-head models, in which all parameters of a generic feature extractor are shared between a set of tasks, and only the weights of the top linear layer (the "heads") are task dependent. In the following, we frame k-shot learning in a similar setting and propose a probabilistic framework for k-shot learning in this vein. Our framework comprises four phases that we refer to as 1) *representational learning*, 2) *concept learning*, 3) *k-shot learning*, and 4) *k-shot testing*, cf. Fig. 1 *(right)*.
We then show that, for certain modelling assumptions, the obtained method is equivalent/related to regularised logistic regression with a specific choice for the regularisation parameter.

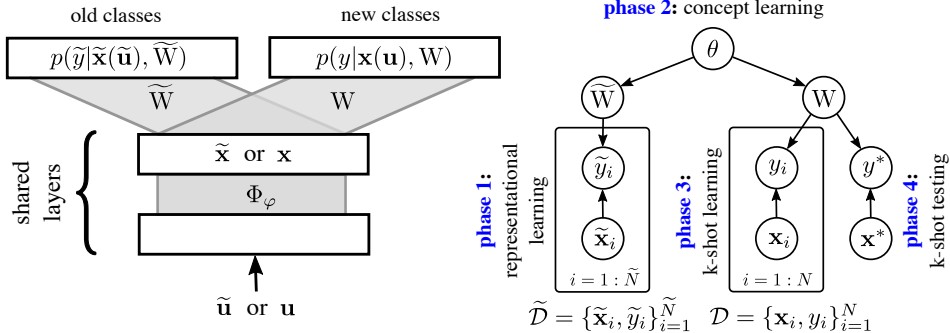

Figure 1: *left:* Shared feature extractor $\Phi_\varphi$ and separate top linear layers W and $\widetilde{W}$ with corresponding softmax units on old and new classes. *right:* Graphical model for probabilistic k-shot learning.

## 2.1 A FRAMEWORK FOR PROBABILISTIC K-SHOT LEARNING

We provide a high-level description of the probabilistic framework and present a more detailed derivation in Appendix A. While it might appear overly formal, the resulting scheme will be simple and practical, and the probabilistic phrasing will make it extensible and automatic (no free parameters).

**Feature extractor and representational learning.** We first introduce a convolutional neural network (CNN) $\Phi_\varphi$ as feature extractor whose last hidden layer activations are mapped to two sets of softmax output units corresponding to the $\widetilde{C}$ classes in the large dataset $\widetilde{\mathcal{D}}$ and the $C$ classes in the small dataset $\mathcal{D}$, respectively. These separate mappings are parametrized by weight matrices $\widetilde{W}$ for the old classes and $W$ for the new classes. Denoting the output of the final hidden layer as $\mathbf{x} = \Phi_\varphi(\mathbf{u})$, the first softmax units compute $p(\widetilde{y}_n \,|\, \widetilde{\mathbf{x}}_n, \widetilde{W}) = \mathrm{softmax}(\widetilde{W}\widetilde{\mathbf{x}}_n)$ and the second $p(y_n \,|\, \mathbf{x}_n, W) = \mathrm{softmax}(W\mathbf{x}_n)$, cf. Fig. 1 *(left)*.

For *representational learning (phase 1)* the large dataset $\widetilde{\mathcal{D}}$ is used to train the CNN $\Phi_\varphi$ using standard deep learning optimisation approaches. This involves learning the parameters $\varphi$ of the feature extractor up to the last hidden layer, as well as the softmax weights $\widetilde{W}$. The network parameters $\varphi$ are fixed from this point on and shared across later phases.

**Probabilistic modelling.** The next goal is to build a probabilistic method for k-shot prediction that transfers structure from the trained softmax weights $\widetilde{W}$ to the new k-shot softmax weights $W$ and combines it with the k-shot training examples. Thus, given a test image $\mathbf{u}^*$ during *k-shot testing (phase 4)*, we compute its feature representation $\mathbf{x}^* = \Phi(\mathbf{u}^*)$, and the prediction for the new label $y^*$ is found by averaging the softmax outputs over the posterior distribution of the softmax weights given the two datasets,

$$p(y^* \,|\, \mathbf{x}^*, \mathcal{D}, \widetilde{\mathcal{D}}) = \int p(y^* \,|\, \mathbf{x}^*, W) p(W \,|\, \mathcal{D}, \widetilde{\mathcal{D}}) \mathrm{d}W. \tag{1}$$

To this end, we consider a general class of probabilistic models in which the two sets of softmax weights are generated from shared hyperparameters $\theta$, so that $p(\widetilde{W}, W, \theta) = p(\theta)p(\widetilde{W}|\theta)p(W|\theta)$ as indicated in the graphical model in Fig. 1 *(right)*. In this way, the large dataset $\widetilde{\mathcal{D}}$ contains information about $\theta$ that in turn constrains the new softmax weights $W$. We further assume that there is very little uncertainty in $\widetilde{W}$ once the large initial training set is observed and so a maximum a posteriori (MAP) estimate, as returned by standard deep learning, suffices. As a consequence of this approximation and the structure of the model, the original data $\widetilde{\mathcal{D}}$ are not required for the k-shot learning phase. Instead, the weights learned from these data, $\widetilde{W}^{\mathrm{MAP}}$, can themselves be treated as observed data, which induce a predictive distribution over the k-shot weights $p(W|\widetilde{W}^{\mathrm{MAP}})$ via Bayes' rule. This argument is fully explained in Appendix A. We refer to this step as *concept learning (phase 2)* and note that all probabilistic modelling happens in the definition of $p(\widetilde{W}, W, \theta)$, (see Secs. 2.2 and 2.3).

During *k-shot learning (phase 3)* we treat this predictive distribution as our new prior on the weights and again use Bayes' rule to combine it with the softmax likelihood of the k-shot training examples $\mathcal{D}$ to obtain a new posterior over the weights that now also incorporates $\mathcal{D}$,

$$p(W \,|\, \mathcal{D}, \widetilde{\mathcal{D}}) \approx p(W \,|\, \mathcal{D}, \widetilde{W}^{\mathrm{MAP}}) \propto p(W \,|\, \widetilde{W}^{\mathrm{MAP}}) \prod_{n=1}^{N} p(y_n|\mathbf{x}_n, W). \tag{2}$$

Finally, we approximate Eq. (2) by its MAP estimate $W^{\mathrm{MAP}}$, so that the integral in Eq. (1) becomes

$$p(y^* \,|\, \mathbf{x}^*, \mathcal{D}, \widetilde{\mathcal{D}}) \approx p(y^* \,|\, \mathbf{x}^*, \mathcal{D}, \widetilde{W}^{\mathrm{MAP}}) \approx p(y^* \,|\, \mathbf{x}^*, W^{\mathrm{MAP}}). \tag{3}$$

## 2.2 CHOOSING A MODEL FOR THE WEIGHTS

The probabilistic model over the weights is key: a good model will transfer useful knowledge that improves performance. However, the usual trade-off between model complexity and learnability is particularly egregious in our setting as the weights $\widetilde{W}$ are few and high-dimensional and the number of k-shot samples is small. With an eye on simplicity, we make two simplifying assumptions. First, treating the weights from the hidden layer to the softmax outputs as a vector, we assume independence.

Second, we assume the distribution between the weights of old and new classes to be identical,

$$p(\widetilde{W}, W, \theta) = p(\theta) \prod_{c'=1}^{\widetilde{C}} p(\widetilde{\mathbf{w}}_{c'} | \theta) \prod_{c=1}^{C} p(\mathbf{w}_c | \theta) \text{ where } p(\widetilde{\mathbf{w}}_{c'} | \theta) \overset{\text{dist}}{=} p(\mathbf{w}_c | \theta). \quad (4)$$

After extensive testing, we found that a Gaussian model for the weights strikes the best compromise in the trade-off between complexity and learnability, cf. Sec. 4.2 for a detailed model comparison.

### 2.3 GAUSSIAN MODEL AND ITS RELATION TO LOGISTIC REGRESSION

**Our method.** We use a simple Gaussian model $p(\mathbf{w} | \theta) = \mathcal{N}(\mathbf{w} | \mu, \Sigma)$ with its conjugate Normal-inverse-Wishart prior $p(\theta) = p(\mu, \Sigma) = \mathcal{NIW}(\mu, \Sigma | \mu_0, \kappa_0, \Lambda_0, \nu_0)$, and estimate MAP solutions for the parameters $\theta^{\text{MAP}} = \{\mu^{\text{MAP}}, \Sigma^{\text{MAP}}\}$. The approximations discussed in Sec. 2.1 lead to $p(W | \widetilde{\mathcal{D}}) \approx p(W | \widetilde{W}^{\text{MAP}}) = \mathcal{N}(W | \mu^{\text{MAP}}, \Sigma^{\text{MAP}})$, and the posterior at k-shot time becomes

$$p(W | \mathcal{D}, \widetilde{\mathcal{D}}) \propto \mathcal{N}(W | \mu^{\text{MAP}}, \Sigma^{\text{MAP}}) \prod_{n=1}^{N} p(y_n | \mathbf{x}_n, W). \quad (5)$$

For details see Appendix C.1. For k-shot testing we use the MAP estimates for the weights of the new classes. We found that restricting the covariance matrix to be isotropic, $\Sigma = \sigma^2 I$, performed best at k-shot learning, probably due to the small number of data points to learn from as mentioned above.

**Relation to logistic regression.** Standard logistic regression corresponds to the maximum likelihood (MLE) solution of the softmax likelihood $p(y_n | \mathbf{x}_n, W) = \text{softmax}(W\mathbf{x}_n)$. Often, $L_2$-regularisation on the weights W with inverse regularisation strength $1/C_{\text{reg}}$ is used; the solution to this regularised optimisation problem corresponds to the MAP solution of a model with isotropic Gaussian prior on the weights with zero mean: $p(W | \mathcal{D}) \propto \mathcal{N}(W | 0, \frac{1}{2} C_{\text{reg}} I) \prod_{n=1}^{N} p(y_n | \mathbf{x}_n, W)$. This method is analogous to Eq. (5). However, the probabilistic framework has several advantages: i) modelling assumptions and approximations are made explicit, ii) it is strictly more general and can incorporate non-zero means $\mu^{\text{MAP}}$, whereas standard regularised logistic regression assumes zero mean, and iii) the probabilistic interpretation provides a principled way of choosing the regularisation constant using the trained weights $\widetilde{W}$: $C_{\text{reg}} = 2\sigma_{\widetilde{W}}^2$, where $\sigma_{\widetilde{W}}^2$ is the empirical variance of the weights $\widetilde{W}^{\text{MAP}}$. In k-shot learning, alternative (frequentist) methods such as cross-validation suffer in the face of the small number of k-shot examples, and are not applicable in 1-shot learning at all.

## 3 RELATED WORK

*Embedding methods* map the k-shot training and test points into a non-linear space and perform classification by assessing which training points are closest, according to a metric, to the test points. Siamese Networks (Koch et al., 2015) train the embedding using a same/different prediction task derived from the original dataset and use a weighted $L_1$ metric for classification. Matching Networks (Vinyals et al., 2016) construct a set of k-shot learning tasks from the original dataset to train an embedding defined through an attention mechanism that linearly combines training labels weighted by their proximity to test points. More recently, Prototypical Networks (Snell et al., 2017) are a streamlined version of Matching Networks in which embedded classes are summarised by their mean in the embedding space. These embedding methods learn representations for k-shot learning, but do not directly leverage concept transfer.

*Amortised optimisation methods* (Ravi & Larochelle, 2017) also simulate related k-shot learning tasks from the initial dataset, but instead train a second network to initialise and optimise a CNN to perform accurate classification on these small datasets. This method can then be applied for new k-shot tasks.

Importantly, both embedding and amortised inference methods improve when the system is trained for a specific $k$-shot task: to perform well in 5-shot learning, training is carried out with episodes containing 5 examples in each class. The general statement appears to be that training specifically for k-shot learning is essential for building features which generalise well at k-shot testing time. The approach proposed in this paper is more flexible; it is not tailored for a specific $k$ and, thus, does not require retraining when switching, e.g., from 5-shot to 10-shot learning. Moreover, Snell et al. (2017)

find that using a larger number of k-shot classes for the training episodes (e.g., train with 20 k-shot classes per episode when testing on only 5 new k-shot classes) can be beneficial, and they choose this number by cross-validation on a validation-set. This is in alignment with our finding that training with more data and more classes improves performance at k-shot time.

*Deep probabilistic methods* include the approach developed in this paper. The methods in this family are not unique to deep learning, and the idea of treating weights as data from which to transfer has been widely applied in multi-task learning (Bakker & Heskes, 2003). The work most closely related to our own is not an approach to k-shot learning *per se*, but rather a method for training CNNs with highly imbalanced classes (Srivastava & Salakhutdinov, 2013). It is similar in that it trains a form of Gaussian mixture model over the final layer weights using MAP inference that regularises learning. Burgess et al. (2016) propose an elegant approach to k-shot learning that is an instance of the framework described here: a Gaussian model is fit to the weights with MAP inference. The evaluation is promising, but preliminary. One of the goals of this paper is to provide a comprehensive evaluation. While not using a probabilistic approach, Qiao et al., 2017 develop a method for k-shot learning that trains a recognition model to amortise MAP inference for the softmax weights which can then be used at k-shot learning time. While this method trains the mapping from activation to weights jointly with the classifier, and thus does not learn from the weights per se, it does exploit the structure in the weights for k-shot learning.

## 4 EXPERIMENTS

The code used to produce the following experiments will be made available after review.

**Dataset.**   *mini*ImageNet has become a standard testbed for k-shot learning and is derived from the ImageNet ILSVRC12 dataset (Russakovsky et al., 2015) by extracting $100$ out of the $1000$ classes. Each class contains $600$ images downscaled to $84 \times 84$ pixels. We use the 100 classes (64 train, 16 validation, 20 test) proposed by Ravi & Larochelle (2017). As our approach does not require a validation set, we use both the training and validation data for the representational learning.

**Representational learning.**   We employ standard CNNs that are inspired by ResNet-34 (He et al., 2016) and VGG (Simonyan & Zisserman, 2014) for the representational learning on the $\widetilde{C}$ base classes, cf. Phase 1 in Sec. 2.1. These trained networks provide both $\widetilde{W}^{\mathrm{MAP}}$ and the fixed feature representation $\Phi_\varphi$ for the k-shot learning and testing. We employed standard data augmentation from ImageNet for the representational learning but highlight that no data augmentation was used during the k-shot training and testing. For details on the architecture, training, and data augmentation see Appendix D.4. t-SNE embeddings (Van der Maaten & Hinton, 2008) of the learned last layer weights show sensible clusters, which highlights the structure exploited by the probabilistic model, see Appendix E.1.

**Baselines and competing methods.**   We compare against several baselines as well as recent state-of-the-art methods mentioned in Sec. 3. The baselines are computed on the features $\mathbf{x} = \Phi_\phi(\mathbf{u})$ from the last hidden layer of the trained CNN: (i) Nearest Neighbours with cosine distance and (ii) regularized logistic regression with regularisation constant set either by cross-validation or (iii) using the variance of the weights, $C = 2\sigma^2_{\widetilde{W}}$, as motivated by our probabilistic framework, cf. Sec. 2.3. We also compare against three recent k-shot methods: (i) Matching Networks[1] (Vinyals et al., 2016), (ii) Prototypical Networks, with numbers reported from Snell et al., 2017 and (iii) Meta-learner LSTM, with numbers reported from Ravi & Larochelle, 2017.

**Testing protocol.**   We evaluate the methods on 600 random k-shot tasks by randomly sampling 5 classes from the 20 test classes and perform 5-way k-shot learning. Following Snell et al. (2017), we use 15 randomly selected images per class for k-shot testing to compute accuracies and calibration.

### 4.1 RESULTS ON *mini*IMAGENET

**Overall k-shot performance.**   We report performance on the *mini*ImageNet dataset in Tab. 1 and Figs. 2 and 3. The best method uses as feature extractor a modified ResNet-34 with 256 features,

---

[1]as reimplemented and optimised by `https://github.com/AntreasAntoniou/MatchingNetworks` to produce results that are superior to those originally published.

| Method | 1-shot | 5-shot | 10-shot |
|---|---|---|---|
| **ResNet-34 + Isotropic Gaussian (ours)** | **$56.3 \pm 0.4\%$** | **$73.9 \pm 0.3\%$** | **$78.5 \pm 0.3\%$** |
| Matching Networks (reimplemented, 1-shot) | $46.8 \pm 0.5\%$ | - | - |
| Matching Networks (reimplemented, 5-shot) | - | $62.7 \pm 0.5\%$ | - |
| Meta-Learner LSTM (Ravi & Larochelle, 2017) | $43.4 \pm 0.8\%$ | $60.6 \pm 0.7\%$ | - |
| Prototypical Nets (1-shot) (Snell et al., 2017) | $49.4 \pm 0.8\%$ | $65.4 \pm 0.7\%$ | - |
| Prototypical Nets (5-shot) (Snell et al., 2017) | $45.1 \pm 0.8\%$ | $68.2 \pm 0.7\%$ | - |

Table 1: Accuracy on 5-way classification on *mini*ImageNet. Our best method, an isotropic Gaussian model using ResNet-34 features consistently outperforms all competing methods by a wide margin.

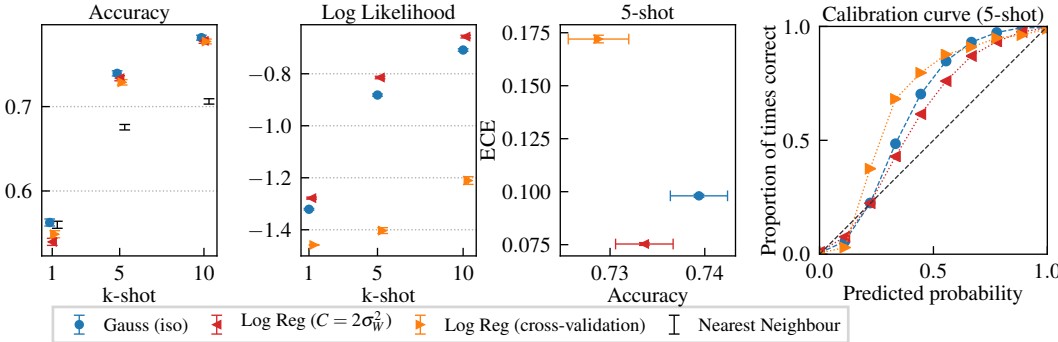

Figure 2: Results for *mini*ImageNet with ResNet-34 style architecture and 600 training images per class. From left to right: accuracy and log likelihood (higher is better) for different k, Expected Calibration Error (ECE, lower is better) vs accuracy for 5-shot learning, and Calibration curve for 5-shot learning. Results on other architectures can be found in Appendix E.2
.

trained with all 600 examples per training class, and a simple isotropic Gaussian model on the weights for concept learning. Despite its simplicity, our method achieves state-of-the-art and beats prototypical networks by a wide margin of about $6\%$. The baseline methods using the same feature extractor are also state-of-the-art compared to prototypical networks and both logistic regressions show comparable accuracy to our methods except for on 1-shot learning. In terms of log-likelihoods, Log Reg ($C = 2\sigma^2_{\widetilde{W}}$) fares slightly better, whereas Log Reg (cv) is much worse.

**Deeper features lead to better k-shot learning.** We investigate the influence of different feature extractors of increasing complexity on performance in Fig. 3: i) a VGG style network (500 train images per class), ii) a ResNet-34 (500 examples per class), and iii) a ResNet-34 (all 600 examples per class). We find that the complexity of the feature extractor as well as training set size consistently correlate with the accuracy at k-shot time. For instance, on 5-shot, Gauss (iso) achieves $65\%$ accuracy with a VGG network and $74\%$ with a ResNet trained with all available data, a significant increase of almost $10\%$. Moreover, Gauss (iso) outperforms Log Reg ($C = 2\sigma^2_{\widetilde{W}}$) on 1-shot learning across models, and performs similarly on 5- and 10-shot. We attribute the difference to the former's ability to also model the mean of the Gaussian, whereas logistic regression assumes a zero mean.

Importantly, this result implies that training specifically for k-shot learning is not necessary for achieving high generalisation performance on this k-shot problem. On the contrary, training a powerful deep feature extractor on batch classification using all of the available training data, then building a simple probabilistic model using the learned features and weights achieves state-of-the-art. Recent models that use episodic training cannot leverage such deep feature extractors as for them the depth of the model is limited by the nature of training itself. The reference baseline in the k-shot learning literature is nearest neighbours, which performs on par with Gauss (iso) on 1-shot learning but is outperformed by all methods on 5- and 10-shot. This is evidence that building a simple classifier on top of the learned features works significantly better for k-shot learning than nearest neighbours.

**Calibration.** A classifier is said to be calibrated when the probability it predicts for belonging to a given class is on par with the probability of it being the correct prediction. In other words, when

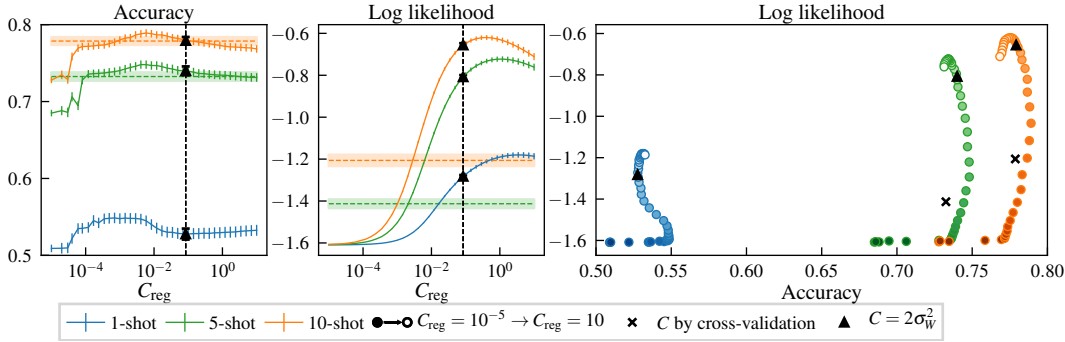

Figure 3: Comparison of different network architectures and training set sizes on the k-shot learning task: VGG style network (trained on 500 images per class) and ResNet-34 style network (trained on 500 and 600 images per class, respectively). Both, deeper networks and larger number of training images, give rise to features that transfer better to k-shot learning.

Figure 4: Choice of regularisation constant for logistic regression for k-shot learning. Results for $C_{\text{reg}} = 2\sigma_{\widetilde{W}}^2$ are drawn as black triangles. Dashed lines correspond to logistic regression with cross-validated (changing) regularisation constant. Colour brightness of the markers ranges from dark ($C = 10^{-5}$) to bright ($C = 10$). ECE plots are provided in Appendix E.3.

examples for which it predicts a probability $p$ of belonging to a given class are correctly classified for a fraction $p$ of the examples. A calibration curve visualises the proportion of examples correctly classified as a function of their predicted probability; a perfectly calibrated classifier should result in a diagonal line. Following Guo et al. (2017), we consider the log likelihood on the k-shot test examples as well as Expected Calibration Error (ECE) as summary measures of calibration. ECE can be interpreted as the weighted average of the distance of the calibration curve to the diagonal. We find that Log Reg ($C = 2\sigma_{\widetilde{W}}^2$) and Gauss (iso) provide better accuracy and calibration than Log Reg (cross-validation), cf. Fig. 2. The difference in calibration quality for different regularisations of logistic regression highlights the importance of choosing the right constant, as we discuss now.

**Choice of the regularisation constant for logistic regression.**   The results so far suggest that training a simple linear model such as regularised logistic regression might be sufficient to perform well in k-shot learning. However, while the accuracy at k-shot time does not vary dramatically as the regularisation constant changes, the calibration does, and jointly maximizing both quantities is not possible, cf. the first two plots of Fig. 4. The standard (frequentist) method to tune this constant is cross validation, which is not applicable in the 1-shot setting, and suffers from lack of data in 5- and 10-shot. Contrary, our probabilistic framework provides a principled way of selecting this regularisation parameter by transfer from the training weights: Log Reg ($C = 2\sigma_{\widetilde{W}}^2$) strikes a good balance between accuracy and log-likelihood. The third plot in Fig. 4 reports log-likelihood as a function of accuracy and provides further visualisation of the achieved trade-off between accuracy and calibration for Log Reg ($C = 2\sigma_{\widetilde{W}}^2$), as well as the failure of Log Reg (cross-validation) to achieve a good compromise in 5- and 10-shot.

**Evaluation in an online setting.**   We also briefly consider the online setting, in which we jointly test on 80 old and 5 new classes, for which catastrophic forgetting (French, 1999) is a well known problem. During k-shot learning and testing we employ a softmax which includes both the new and the old weights resulting in a total of 85 weight vectors. We utilise ResNet-34 trained on 500 images per class to retain 100 test images on the old classes. While the k-shot weights were modelled probabilistically, we use the MAP estimate $\widetilde{W}^{\text{MAP}}$ for the old weights. Accuracies are reported in

Figure 5: Online learning with ResNet-34 features. Gauss (iso) and Log Reg ($2\sigma^2_{\widetilde{W}}$) strike a good trade-off between learning on new classes and forgetting of old classes. Unregularised Log Reg (MLE) and Log Reg ($2\sigma^2_{\widetilde{W}}$, only new), which has not been trained in the presence of the old weights, either completely forget the old classes or do not learn anything, respectively.

Fig. 5 for i) all the 85 classes, ii) the old 80 classes only, and iii) the new 5 classes only. For 5- and 10-shot, Gauss (iso) and Log Reg ($2\sigma^2_{\widetilde{W}}$) only lose a couple of percent on the accuracy of the old classes, and perform well on the new classes, striking a good trade-off between forgetting and learning at k-shot time. For unregularised (MLE) logistic regression, the new weights completely dominate the old ones, highlighting that the right regularisation is important. Yet, cross-validation in this setting is often very challenging. When training Logistic Regression without including the old weights ("only new"), the new weights are dominated by the old ones and fail to learn the new classes, making *training in the presence of the old weights* an essential component for online learning.

## 4.2 MODEL COMPARISON ON CIFAR-100

We performed an extensive comparison between different probabilistic models of the weights using different inference procedures, which we present in Appendix E.4. We report results on the CIFAR-100 dataset on (i) Gaussian, (ii) mixture of Gaussians, and (iii) Laplace, all with either MAP estimation or Hybrid Monte Carlo sampling. We found that the simple Gaussian model is on par with or outperforms other methods at k-shot time, which we attribute to it striking a good balance between choosing a complex model, which may better fit the weights, and statistical efficiency, as the number of weights $\widetilde{C}$ (80 in our case) is often smaller than the dimensionality of the feature representation (256 in our case), cf. Sec. 2. This finding is supported by computing the log-likelihood of held out training weights under such model, with the Gaussian model performing best. Experiments using Hybrid Monte Carlo sampling for k-shot learning returned very similar performance to MAP estimation and at a much higher computational cost, due to the difficulty of performing sampling in such a high dimensional parameter space. Our recommendation is that practitioners should use simple models and employ simple inference schemes to estimate all free parameters thereby avoiding expending valuable data on validation sets.

## 5 CONCLUSION

We present a probabilistic framework for k-shot learning that exploits the powerful features and class information learned by a neural network on a large training dataset. Probabilistic models are then used to transfer information in the network weights to new classes. Experiments on *mini*ImageNet using a simple Gaussian model within our framework achieve state-of-the-art for 1-shot and 5-shot learning by a wide margin, and at the same time return well calibrated predictions. This finding is contrary to the current belief that episodic training is necessary to learn good k-shot features and puts the success of recent complex deep learning approaches to k-shot learning into context. The new approach is flexible and extensible, being applicable to general discriminative models and k-shot learning paradigms. For example, preliminary results on online k-shot learning indicate that the probabilistic framework mitigates catastrophic forgetting by automatically balancing performance on the new and old classes.

The Gaussian model is closely related to regularised logistic regression, but provides a principled and fully automatic way to regularise. This is particularly important in k-shot learning, as it is a low-data regime, in which cross-validation performs poorly and where it is important to train on all available data, rather than using validation sets.

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

# Appendix to "Discriminative k-shot learning using probabilistic models"

## A   DETAILS ON THE DERIVATION AND APPROXIMATIONS FROM SEC. 2.1

As stated in the main text, the probabilistic k-shot learning approach comprises four phases mirroring the dataflow:

**Phase 1: Representational learning.**   The large dataset $\widetilde{\mathcal{D}}$ is used to train the CNN $\Phi_\varphi$ using standard deep learning optimisation approaches. This involves learning both the parameters $\varphi$ of the feature extractor up to the last hidden layer, as well as the softmax weights $\widetilde{W}$. The network parameters $\varphi$ are fixed from this point on and shared across phases. This is a standard setup for multi-task learning and in the present case it ensures that the features derived from the representational learning can be leveraged for k-shot learning.

**Phase 2: Concept learning.**   The softmax weights $\widetilde{W}$ are effectively used as data for concept learning by training a probabilistic model that detects structure in these weights which can be transferred for k-shot learning. This approach will be justified in the next section. For the moment, we consider a general class of probabilistic models in which the two sets of weights are generated from shared hyperparameters $\theta$, so that $p(\widetilde{W}, W, \theta) = p(\theta)p(\widetilde{W}|\theta)p(W|\theta)$ (see Fig. 1).

**Phases 3 and 4: k-shot learning and testing.**   Probabilistic k-shot learning leverages the learned representation $\Phi_\varphi$ from phase 1 and the probabilistic model $p(\widetilde{W}, W, \theta)$ from phase 2 to build a (posterior) predictive model for unseen new examples using examples from the small dataset $\mathcal{D}$.

### PROBABILISTIC MODEL OF THE WEIGHTS

Given the dataflow and the assumed probabilistic model in Fig. 1 *(right)*, a completely probabilistic approach would involve the following steps.

In the concept learning phase, the initial dataset would be used to form the posterior distribution over the concept hyperparameters $p(\theta \,|\, \widetilde{\mathcal{D}})$. The k-shot learning phase combines the information about the new weights provided by $\widetilde{\mathcal{D}}$ with the information in the k-shot dataset $\mathcal{D}$ to form the posterior distribution

$$p(W \,|\, \mathcal{D}, \widetilde{\mathcal{D}}) \propto p(W \,|\, \widetilde{\mathcal{D}}) \prod_n p(y_n \,|\, \mathbf{x}_n, W) \ \text{ where } \ p(W \,|\, \widetilde{\mathcal{D}}) = \int p(W \,|\, \theta)p(\theta \,|\, \widetilde{\mathcal{D}})\mathrm{d}\theta. \quad (6)$$

To see this, notice that

$$p(W \,|\, \mathcal{D}, \widetilde{\mathcal{D}}) \propto p(W, \mathcal{D}, \widetilde{\mathcal{D}}) = p(\widetilde{\mathcal{D}})p(W \,|\, \widetilde{\mathcal{D}})p(\mathcal{D} \,|\, \widetilde{\mathcal{D}}, W). \quad (7)$$

The graphical model in Fig. 1 entails that $\mathcal{D}$ is conditionally independent from $\widetilde{\mathcal{D}}$ given $W$, such that

$$p(\mathcal{D} \,|\, W, \widetilde{\mathcal{D}}) = p(\mathcal{D} \,|\, W) = \prod_n p(y_n \,|\, x_n, W). \quad (8)$$

We recover Eq. (6) by adding $p(\widetilde{\mathcal{D}})$ to the constant of proportionality.

Inference in this model is generally intractable and requires approximations. The main challenge is computing the posterior distribution over the hyper-parameters given the initial dataset. However, progress can be made if we assume that the posterior distribution over the weights can be well approximated by the MAP value $p(\widetilde{W} \,|\, \widetilde{\mathcal{D}}) \approx \delta(\widetilde{W} - \widetilde{W}^{\mathrm{MAP}})$. This is an arguably justifiable assumption as the initial dataset is large and so the posterior will concentrate on narrow modes (with similar predictive performance). In this case $p(\theta \,|\, \widetilde{\mathcal{D}}) \approx p(\theta \,|\, \widetilde{W}^{\mathrm{MAP}})$ and, due to the structure of the probabilistic model, all instances of $\widetilde{\mathcal{D}}$ in Eq. (6) and Eq. (1) can be replaced by the analogous expressions involving $\widetilde{W}^{\mathrm{MAP}}$. This greatly simplifies the learning pipeline as the probabilistic modelling only needs to have access to the weights returned by representational learning. Remaining intractabilities involve only a small number of data points $\mathcal{D}$ and can be handled using standard approximate inference tools. The following summarizes the approximations and computational steps for each phase of training.

**Phase 1: Representational learning.** Deep learning is used to train a CNN. The representation of input images at the last hidden layer, $\mathbf{x} = \Phi_\varphi(\mathbf{u})$, is used in subsequent phases. The final layer softmax weights are assumed to be MAP estimates $\widetilde{\mathrm{W}}^{\mathrm{MAP}}$.

**Phase 2: Concept learning.** A probabilistic model is fit directly to the MAP weights $p(\theta \,|\, \widetilde{\mathrm{W}}^{\mathrm{MAP}}) \propto p(\theta)p(\widetilde{\mathrm{W}}^{\mathrm{MAP}}|\theta)$. For conjugate models a full posterior can be retained, otherwise a MAP estimate $p(\theta \,|\, \widetilde{\mathrm{W}}^{\mathrm{MAP}}) \approx \delta(\theta - \theta^{\mathrm{MAP}})$ is used.

**Phase 3: k-shot learning.** The posterior distribution over the new softmax weights $p(\mathrm{W} \,|\, \mathcal{D}, \widetilde{\mathrm{W}}^{\mathrm{MAP}}) \propto p(\mathrm{W} \,|\, \widetilde{\mathrm{W}}^{\mathrm{MAP}}) \prod_{n=1}^{N} p(y_n|\mathbf{x}_n, \mathrm{W})$ is generally intractable. The posterior can, however, be approximated using the MAP estimate $p(\mathrm{W} \,|\, \mathcal{D}, \widetilde{\mathrm{W}}^{\mathrm{MAP}}) \approx \delta(\mathrm{W} - \mathrm{W}^{\mathrm{MAP}})$ or through sampling $\mathrm{W}_m \sim p(\mathrm{W} \,|\, \mathcal{D}, \widetilde{\mathrm{W}}^{\mathrm{MAP}})$. Note that $p(\mathrm{W} \,|\, \widetilde{\mathrm{W}}^{\mathrm{MAP}}) = \int p(\mathrm{W}|\theta)p(\theta \,|\, \widetilde{\mathrm{W}}^{\mathrm{MAP}})\mathrm{d}\theta$ is analytic for conjugate models and, if instead a MAP estimate for $\theta$ is provided by the concept modelling stage, then $p(\mathrm{W} \,|\, \widetilde{\mathrm{W}}^{\mathrm{MAP}}) \approx p(\mathrm{W}|\theta^{\mathrm{MAP}})$.

**Phase 4: k-shot testing.** Approximate inference is used to compute $p(y^* \,|\, \mathbf{x}^*, \mathcal{D}, \widetilde{\mathrm{W}}^{\mathrm{MAP}}) = \int p(y^* \,|\, \mathbf{x}^*, \mathrm{W})p(\mathrm{W} \,|\, \mathcal{D}, \widetilde{\mathrm{W}}^{\mathrm{MAP}})\mathrm{dW}$. If the k-shot learning phase provides a MAP estimate of W then $p(y^* \,|\, \mathbf{x}^*, \mathcal{D}, \widetilde{\mathrm{W}}^{\mathrm{MAP}}) \approx p(y^* \,|\, \mathbf{x}^*, \mathrm{W}^{\mathrm{MAP}})$. If samples are returned then $p(y^* \,|\, \mathbf{x}^*, \mathcal{D}, \widetilde{\mathrm{W}}^{\mathrm{MAP}}) \approx \frac{1}{M} \sum_{m=1}^{M} p(y^* \,|\, \mathbf{x}^*, \mathrm{W}_m)$.

## B  APPROXIMATE INFERENCE METHODS

In this section we briefly discuss different inference methods for the probabilistic models. In the main text we only considered MAP inference as we found that other more complicated inference schemes do not yield a practical benefit. However, in Appendix E.4 we provide a detailed model comparison, in which we also consider other approximate inference methods.

In all cases the gradients of the densities w.r.t. W can be computed, enabling MAP inference in the k-shot learning phase to be efficiently performed via gradient-based optimisation using L-BFGS (Liu & Nocedal, 1989). Alternatively, Markov Chain Monte Carlo (MCMC) sampling can be performed to approximate the associated integral, see Eq. (1). Due to the high dimensionality of the space and as gradients are available, we employ Hybrid Monte Carlo (HMC) (Neal et al., 2011) sampling in the form of the recently proposed NUTS sampler that automatically tunes the HMC parameters (step size and number of leapfrog steps) (Hoffman & Gelman, 2014). For the GMMs we employed `pymc3` (Salvatier et al., 2016) to perform MAP inference.

## C  MODELS FOR THE PRIOR ON THE WEIGHTS

As discussed in Sec. 2.1, we specify our model through $p(\mathrm{W}, \widetilde{\mathrm{W}}, \theta)$ thus defining $p(\mathrm{W} \,|\, \widetilde{\mathrm{W}}^{MAP})$ in Eq. (2). This section analyses different priors on the weights: (i) Gaussian models, (ii) Gaussian mixture models, and (iii) Laplace distribution. In the main paper, we only use a Gaussian model with MAP inference, as we saw no significant advantage in using other, more complex models. However, we provide an extensive comparison of the different models in Appendix E.4.

### C.1  GAUSSIAN MODEL

Possibly the simplest approach consists of modelling $p(\mathrm{W} \,|\, \widetilde{\mathrm{W}})$ as a Gaussian distribution:

$$p(\mathrm{W} \,|\, \widetilde{\mathrm{W}}) = \int \mathcal{N}(\mathrm{W} \,|\, \mu, \Sigma)p(\mu, \Sigma \,|\, \widetilde{\mathrm{W}})\mathrm{d}\mu\mathrm{d}\Sigma. \tag{9}$$

Details for this section can be found in Murphy, 2012. The normal-inverse-Wishart distribution for $\mu$ and $\Sigma$ is a conjugate prior for the Gaussian, which allows for the posterior to be written in closed form. More precisely,

$$\begin{aligned}
p(\mu, \Sigma) &= \mathcal{NIW}(\mu, \Sigma \,|\, \mu_0, \kappa_0, \Lambda_0, \nu_0) \\
&= \frac{1}{Z}|\Sigma|^{-(\nu_0+p)/2+1}e^{-\frac{1}{2}tr(\Lambda_0\Sigma^{-1})-\frac{\kappa_0}{2}(\mu-\mu_0)^t\Sigma^{-1}(\mu-\mu_0)},
\end{aligned} \tag{10}$$

where $Z$ is the normalising constant. The posterior $p(\mu, \Sigma \,|\, \widetilde{\mathrm{W}})$ also follows a normal-inverse-Wishart distribution:

$$p(\mu, \Sigma \,|\, \widetilde{\mathrm{W}}) = \mathcal{NIW}(\mu, \Sigma \,|\, \mu_{\widetilde{N}}, \kappa_{\widetilde{N}}, \Lambda_{\widetilde{N}}, \nu_{\widetilde{N}}), \tag{11}$$

where

$$\mu_{\widetilde{N}} = \frac{\kappa_0}{\kappa_0 + \widetilde{N}}\mu_0 + \frac{\widetilde{N}}{\kappa_0 + \widetilde{N}}\overline{\overline{\widetilde{\mathrm{W}}}}$$

$$\kappa_{\widetilde{N}} = \kappa_0 + \widetilde{N}$$

$$\Lambda_{\widetilde{N}} = \Lambda_0 + S + \frac{\kappa_0 \widetilde{N}}{\kappa_0 + \widetilde{N}}(\overline{\overline{\widetilde{\mathrm{W}}}} - \mu_0)(\overline{\overline{\widetilde{\mathrm{W}}}} - \mu_0)^t$$

$$\nu_{\widetilde{N}} = \nu_0 + \widetilde{N},$$

and $S$ is the sample covariance of $\widetilde{\mathrm{W}}$.

For this model, we can integrate (9) in closed form, which results in the following multivariate Student $t$-distribution:

$$p(\mathrm{W} \,|\, \widetilde{\mathrm{W}}) = t_{\nu_{\widetilde{N}} - p + 1}\left(\mu_{\widetilde{N}}, \frac{\Lambda_{\widetilde{N}}(\kappa_{\widetilde{N}} + 1)}{\kappa_{\widetilde{N}}(\nu_{\widetilde{N}} - p + 1)}\right).$$

As with other approaches, one can also compute the MAP solutions for the mean $\mu_{\mathrm{MAP}}$ and covariance $\Sigma_{\mathrm{MAP}}$, such that $p(\mathrm{W} \,|\, \widetilde{\mathrm{W}}) = \mathcal{N}(\mathrm{W} \,|\, \mu_{\mathrm{MAP}}, \Sigma_{\mathrm{MAP}})$.

For both the analytic posterior and the MAP approximation, $p(\mathrm{W} \,|\, \widetilde{\mathrm{W}})$ depends on the hyperparameters of the normal-inverse-Wishart distribution: $\mu_0, \nu_0, \kappa_0$ and $\Lambda_0$. There are different ways to choose these hyperparameters. One way would be by optimising the log probability of held out training weights, see Appendix E.4 for a brief discussion. In practise, it is common to choose uninformative or data dependent priors as discussed by Murphy (2012, Chapter 4).

## C.2 MIXTURE OF GAUSSIANS (GMM)

A Gaussian mixture model can potentially leverage cluster structure in the weights (animal classes might have similar weights, for example). This is related to the tree-based prior proposed in Srivastava & Salakhutdinov (2013). MAP inference is performed because exact inference is intractable. Similarly to the Gaussian case, different structures for the covariance of each cluster were tested. In our experiments, we fit the parameters of the GMM via maximum likelihood using the EM algorithm. GMM consists on modelling $p(\mathrm{W} \,|\, \widetilde{\mathrm{W}})$ as a mixture of Gaussians with $S$ components:

$$p(\mathrm{W} \,|\, \widetilde{\mathrm{W}}) = \int \left(\sum_{s=1}^{S} \pi_s \mathcal{N}(\mathrm{W} \,|\, \mu_s, \Sigma_s)\right) p(\mu_1, \ldots, \mu_S, \Sigma_1, \ldots, \Sigma_S \,|\, \widetilde{\mathrm{W}}) \mathrm{d}\mu_1 \ldots \mathrm{d}\mu_S \mathrm{d}\Sigma_1 \ldots \mathrm{d}\Sigma_S, \tag{12}$$

where $\sum_{s=1}^{S} \pi_s = 1$. In this work, we only compute the MAP mean and covariance for each of the clusters, as opposed to averaging over the parameters of the mixture. The resulting posterior is

$$p(\mathrm{W} \,|\, \widetilde{\mathrm{W}}) = \sum_{s=1}^{S} \pi_s \mathcal{N}(\mathrm{W} \,|\, \mu_{MAP,s}, \Sigma_{MAP,s}). \tag{13}$$

The components of the mixture are fit in two ways. For CIFAR-100, the classes are grouped into 20 superclasses, each containing 5 of the 100 classes. One option is therefore to initialize 20 components, each fit with the data points in the corresponding superclass. For each such individual Gaussian, the MAP inference method presented in the previous section can be used. In order to increase the number of weight examples in each superclass, we merge the original superclasses into 9 larger superclasses. The merging of the superclasses is the following:

- Aquatic mammals + fish

- flowers + fruit and vegetables + trees

- insects + non-insect invertebrates + reptiles

- medium-sized mammals + small mammals

- large carnivores + large omnivores and herbivores

- people

- large man-made outdoor things + large natural outdoor things

- food containers + household electrical devices + household furniture

- Vehicles 1 + Vehicles 2.

The parameters of the mixture can also be fit using maximum likelihood with EM. We use the implementation of EM in scikit-learn. Both 3 and 10 clusters are considered in CIFAR-100. Weight log-likelihoods under this model and k-shot performance can be found in Appendix E.4.

Note that, similarly to the Gaussian model, we consider isotropic, diagonal or full covariance models for the covariance matrices.

### C.3 LAPLACE DISTRIBUTION

Sparsity is an attractive feature which could be helpful for modelling the weights. Indeed, it is reasonable to assume that each class uses a set of characteristic features which drive classification accuracy, while others are irrelevant. Sparse models would then provide sensible regularization. As such, we consider a product of independent Laplace distribution. Sec. 2.3 highlights the relation between a Gaussian prior on the weights and $L_2$ regularised logistic regression. One can similarly show that the Laplace prior is related to $L_1$ regularised logistic regression, which is well known for encouraging sparse weight vectors.

We consider a prior which factors along the feature dimensions:

$$p(\widetilde{\mathrm{W}} \,|\, \{\mu_j\}, \{\lambda_j\}) = \prod_j^p \frac{1}{2\lambda_j} \exp\left(-\sum_i^{\widetilde{C}} \frac{|\widetilde{\mathrm{W}}_{ij} - \mu_j|}{\lambda_j}\right).$$

where the product over $j$ is along the feature dimensions and the sum over $i$ is across the classes. We fit the parameters $\mu$ and $\lambda$ via maximum likelihood:

$$\mu_{\mathrm{MLE},j} = \mathrm{median}_i(\widetilde{\mathrm{W}}_{ij})$$

$$\lambda_{\mathrm{MLE},j} = \frac{1}{N} \sum_i |\widetilde{\mathrm{W}}_{ij} - \mu_j|,$$

such that

$$p(\mathrm{W} \,|\, \widetilde{\mathrm{W}}) = \prod_j^p \frac{1}{2\lambda_{\mathrm{MLE},j}} \exp\left(-\sum_i^C \frac{|\mathrm{W}_{ij} - \mu_{\mathrm{MLE},j}|}{\lambda_{\mathrm{MLE},j}}\right).$$

An isotropic Laplace model with mean $\mu$ and scale $\lambda$ is also considered:

$$p(\widetilde{\mathrm{W}} \,|\, \mu, \lambda) = \frac{1}{2\lambda} \exp\left(-\frac{\sum_{ij} |\widetilde{\mathrm{W}}_{ij} - \mu|}{\lambda}\right),$$

where

$$\mu_{\mathrm{MLE}} = \mathrm{median}(\widetilde{\mathrm{W}})$$

$$\lambda_{\mathrm{MLE}} = \frac{1}{Np} \sum_{ij} |\widetilde{\mathrm{W}}_{ij} - \mu|,$$

## D    TRAINING AND EVALUATION PROCEDURE DETAILS

### D.1    *mini*IMAGENET

To construct *mini*ImageNet we use the same classes as initially proposed by Ravi & Larochelle (2017) and used in (Snell et al., 2017), which is split into 64 training classes (cf. Tab. 2), 16 validation classes (cf. Tab. 3), and 20 test classes (cf. Tab. 4). We will make a full list of image files available.

As we do not require a validation set, we combine the training and validation set to form an extended training set. We extract 600 images per class from the ImageNet 2012 Challange dataset (Krizhevsky et al., 2012), scale the shorter side to $84$ pixels and then centrally crop to $84 \times 84$ pixels, that is, we preserve the original aspect ratio of the image content. We use these coloured $84 \times 84 \times 3$ images as input for representational and k-shot learning and testing.

In order to train very deep models, such as a ResNet, we need to perform data augmentation as is the case when training full ImageNet. We use the following standard data augmentation from ImageNet that we adapt to the size of the input images:

- random horizontal flipping
- randomly paste image into $100 \times 100$ frame and cut out central $84 \times 84$ pixels
- randomly change brightness, contrast, saturation and lighting

We highlight that we do not perform any data augmentation for the k-shot learning and k-shot testing but use the original $84 \times 84$ colour images as input to the feature extractor.

### D.2    CIFAR-100

CIFAR-100 consists of $100$ classes each with $500$ training and $100$ test images of size $32 \times 32$. The classes are grouped into 20 superclasses with 5 classes each. For example, the superclass "fish" contains the classes aquarium fish, flatfish, ray, shark, and trout. Unless otherwise stated, we used a random split into 80 base classes and 20 k-shot learning classes.

For k-shot learning and testing, we split the 100 classes into 80 base classes used for network training and 20 k-shot learning classes.

```
classes_base = [
    0, 1, 2, 3, 4, 5, 6, 7, 9, 10, 13, 14, 15, 16, 17, 18, 19, 21, 22,
    24, 25, 27, 28, 32, 34, 35, 36, 38, 40, 42, 43, 44, 45, 46, 48, 49,
    50, 51, 52, 53, 54, 55, 56, 58, 59, 60, 61, 62, 63, 64, 65, 66, 67,
    69, 70, 73, 74, 75, 76, 77, 78, 79, 80, 82, 83, 85, 86, 87, 88, 89,
    90, 91, 92, 93, 94, 95, 96, 97, 98, 99
]
classes_heldout = [
    8, 11, 12, 20, 23, 26, 29, 30, 31, 33, 37, 39, 41, 47, 57, 68, 71,
    72, 81, 84
]
```

We provide an exhaustive comparison of different probabilistic models for this k-shot learning task in Appendix E.4.

### D.3    NETWORK ARCHITECTURE AND TRAINING: RESNET INSPIRED

The network architecture is inspired by the ResNet-34 architecture for ImageNet (He et al., 2016) that uses convolution blocks, with two convolutions each, that are bridged by skip connections. As a base, we utilise the example code[2] provided by `tensorpack` (https://github.com/ppwwyyxx/tensorpack), a neural network training library built on top of `tensorflow` (Martín Abadi et al., 2015). We adapt the number of features as well as the size of the last fully connected layer to account for the smaller number of training samples and training classes. The final architecture is detailed in Tab. 5.

---

[2]https://github.com/ppwwyyxx/tensorpack/tree/master/examples/ResNet

| | |
|---|---|
| n03400231 | frying pan, frypan, skillet |
| n02108551 | Tibetan mastiff |
| n02687172 | aircraft carrier, carrier, flattop, attack aircraft carrier |
| n04296562 | stage |
| n13133613 | ear, spike, capitulum |
| n02165456 | ladybug, ladybeetle, lady beetle, ladybird, ladybird beetle |
| n03337140 | file, file cabinet, filing cabinet |
| n02966193 | carousel, carrousel, merry-go-round, roundabout, whirligig |
| n02074367 | dugong, Dugong dugon |
| n02105505 | komondor |
| n04389033 | tank, army tank, armored combat vehicle, armoured combat vehicle |
| n09246464 | cliff, drop, drop-off |
| n03924679 | photocopier |
| n03527444 | holster |
| n04612504 | yawl |
| n01749939 | green mamba |
| n04251144 | snorkel |
| n03347037 | fire screen, fireguard |
| n04067472 | reel |
| n03998194 | prayer rug, prayer mat |
| n13054560 | bolete |
| n02747177 | ashcan, trash can, garbage can, wastebin, ash bin, ash-bin, ashbin, dustbin, trash barrel, trash bin |
| n04435653 | tile roof |
| n02108089 | boxer |
| n03908618 | pencil box, pencil case |
| n01770081 | harvestman, daddy longlegs, Phalangium opilio |
| n03676483 | lipstick, lip rouge |
| n03220513 | dome |
| n04515003 | upright, upright piano |
| n04258138 | solar dish, solar collector, solar furnace |
| n04509417 | unicycle, monocycle |
| n01704323 | triceratops |
| n04443257 | tobacco shop, tobacconist shop, tobacconist |
| n02089867 | Walker hound, Walker foxhound |
| n01910747 | jellyfish |
| n02111277 | Newfoundland, Newfoundland dog |
| n04243546 | slot, one-armed bandit |
| n01558993 | robin, American robin, Turdus migratorius |
| n03047690 | clog, geta, patten, sabot |
| n03854065 | organ, pipe organ |
| n03476684 | hair slide |
| n02113712 | miniature poodle |
| n07747607 | orange |
| n03838899 | oboe, hautboy, hautbois |
| n07584110 | consomme |
| n02795169 | barrel, cask |
| n03017168 | chime, bell, gong |
| n04275548 | spider web, spider's web |
| n04604644 | worm fence, snake fence, snake-rail fence, Virginia fence |
| n02606052 | rock beauty, Holocanthus tricolor |
| n01843383 | toucan |
| n02457408 | three-toed sloth, ai, Bradypus tridactylus |
| n03062245 | cocktail shaker |
| n03207743 | dishrag, dishcloth |
| n02108915 | French bulldog |
| n06794110 | street sign |
| n02823428 | beer bottle |
| n03888605 | parallel bars, bars |
| n04596742 | wok |
| n02091831 | Saluki, gazelle hound |
| n02101006 | Gordon setter |
| n02120079 | Arctic fox, white fox, Alopex lagopus |
| n01532829 | house finch, linnet, Carpodacus mexicanus |
| n07697537 | hotdog, hot dog, red hot |

Table 2: Training classes for *mini*ImageNet as proposed by Ravi & Larochelle (2017)

n03075370    combination lock
n02971356    carton
n03980874    poncho
n02114548    white wolf, Arctic wolf, Canis lupus tundrarum
n03535780    horizontal bar, high bar
n03584254    iPod
n02981792    catamaran
n03417042    garbage truck, dustcart
n03770439    miniskirt, mini
n02091244    Ibizan hound, Ibizan Podenco
n02174001    rhinoceros beetle
n09256479    coral reef
n02950826    cannon
n01855672    goose
n02138441    meerkat, mierkat
n03773504    missiles

Table 3: Validation classes for *mini*ImageNet as proposed by Ravi & Larochelle (2017)

n02116738    African hunting dog, hyena dog, Cape hunting dog, Lycaon pictus
n02110063    malamute, malemute, Alaskan malamute
n02443484    black-footed ferret, ferret, Mustela nigripes
n03146219    cuirass
n03775546    mixing bowl
n03544143    hourglass
n04149813    scoreboard
n03127925    crate
n04418357    theater curtain, theatre curtain
n02099601    golden retriever
n02219486    ant, emmet, pismire
n03272010    electric guitar
n04146614    school bus
n02129165    lion, king of beasts, Panthera leo
n04522168    vase
n07613480    trifle
n02871525    bookshop, bookstore, bookstall
n01981276    king crab, Alaska crab, Alaskan king crab, Alaska king crab, Paralithodes camtschatica
n02110341    dalmatian, coach dog, carriage dog
n01930112    nematode, nematode worm, roundworm

Table 4: Test classes for *mini*ImageNet as proposed by Ravi & Larochelle (2017)

**ResNet-34 inspired for _mini_ImageNet**

| Output size | Layers |
|---|---|
| $84 \times 84 \times 3$ | Input patch |
| $42 \times 42 \times 32$ | $5 \times 5, 32$, stride 2 |
| $42 \times 42 \times 32$ | $\begin{bmatrix} 3 \times 3, 32 \\ 3 \times 3, 32 \end{bmatrix} \times 3$ |
| $21 \times 21 \times 64$ | $\begin{bmatrix} 3 \times 3, 64 \\ 3 \times 3, 64 \end{bmatrix} \times 4$ |
| $11 \times 11 \times 128$ | $\begin{bmatrix} 3 \times 3, 128 \\ 3 \times 3, 128 \end{bmatrix} \times 6$ |
| $6 \times 6 \times 256$ | $\begin{bmatrix} 3 \times 3, 256 \\ 3 \times 3, 256 \end{bmatrix} \times 3$ |
| 256 | global average pooling |
| $\widetilde{C}$ | fully connected, softmax |

Table 5: Network architecture. All unnamed layers are 2D convolutions with stated kernel size and padding SAME; the output of the shaded layer corresponds to $\Phi_\varphi(\mathbf{u})$, the feature space representation of the image $\mathbf{u}$, which is used as input for probabilistic k-shot learning.

The network is trained using a decaying learning rate schedule and momentum SGD and is implemented in `tensorpack` using `tensorflow`.

## D.4 NETWORK ARCHITECTURE AND TRAINING: VGG INSPIRED

**VGG-style Network for CIFAR-100**

| Output size | Layers |
|---|---|
| $32 \times 32 \times 3$ | Input patch |
| $16 \times 16 \times 64$ | 2× (Conv2D, ELU), Pool |
| $8 \times 8 \times 64$ | 2× (Conv2D, ELU), Pool |
| $4 \times 4 \times 128$ | 2× (Conv2D, ELU), Pool |
| $2 \times 2 \times 128$ | 2× (Conv2D, ELU), Pool |
| $2 \times 2 \times 128$ | Dropout (0.5) |
| 256 | FullyConnected, ELU |
| 256 | Dropout (0.5) |
| 128 | FullyConnected, ELU |
| $\widetilde{C}$ | FullyConnected, SoftMax |

**VGG-style Network for _mini_ImageNet**

| Output size | Layers |
|---|---|
| $84 \times 84 \times 3$ | Input patch |
| $42 \times 42 \times 32$ | 2× (Conv2D, ELU), Pool |
| $21 \times 21 \times 64$ | 2× (Conv2D, ELU), Pool |
| $11 \times 11 \times 128$ | 2× (Conv2D, ELU), Pool |
| $6 \times 6 \times 128$ | 2× (Conv2D, ELU), Pool |
| $3 \times 3 \times 128$ | 2× (Conv2D, ELU), Pool |
| $3 \times 3 \times 128$ | Dropout (0.5) |
| 512 | FullyConnected, ELU |
| 512 | Dropout (0.5) |
| 256 | FullyConnected, ELU |
| $\widetilde{C}$ | FullyConnected, SoftMax |

Table 6: Network architectures. All 2D convolutions have kernel size $3 \times 3$ and padding SAME; max-pooling is performed with stride 2. The output of the shaded layer corresponds to $\Phi_\varphi(u)$, the feature space representation of the image $u$, which is used as input for probabilistic k-shot learning

The network architecture was inspired by the VGG networks Simonyan & Zisserman, 2014, but does not employ batch normalisation Ioffe & Szegedy, 2015. To speed up training, we employ exponential linear units (ELUs), which have been reported to lead to faster convergence as compared to ordinary ReLUs Clevert et al., 2015. To regularise the networks, we employ dropout (Srivastava, Hinton, et al., 2014) and regularisation of the weights in the fully connected layers. The networks are trained with the ADAM optimiser Kingma & Ba, 2014 with decaying learning rate.

The network is implemented in `tensorpack` using `tensorflow`.

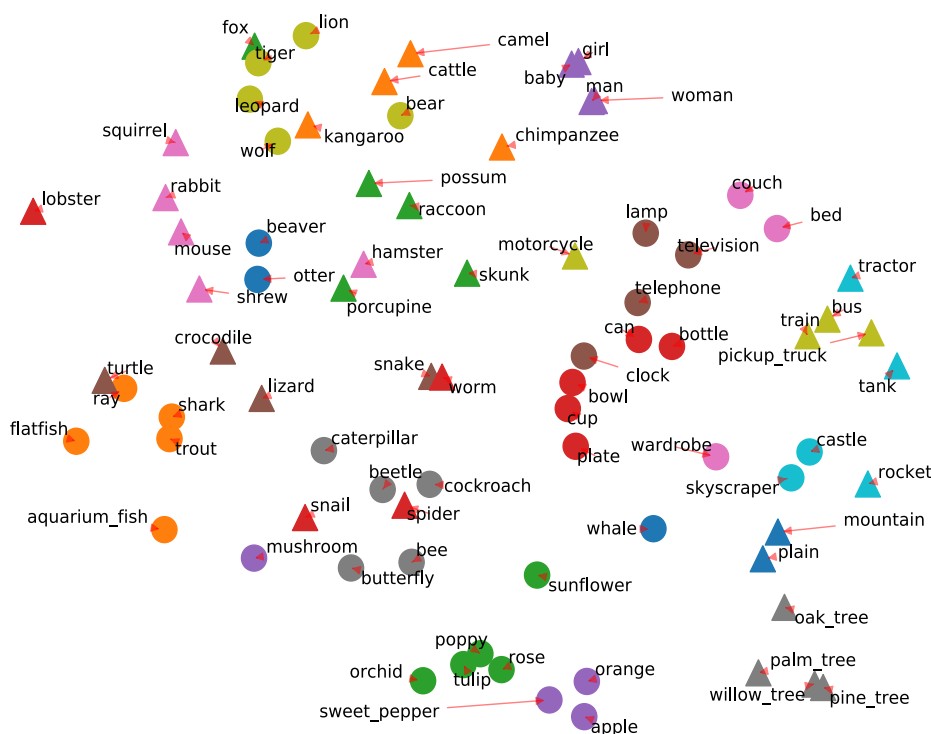

Figure 6: t-SNE embedding of the CIFAR-100 weights $\widetilde{W}$ trained using a VGG style architecture. The points are coloured according to their respective superclass. The colouring by superclass makes the structure in the weights evident, as t-SNE overall recovers the structure in the dataset. For instance, oak tree, palm tree, willow tree and pine tree form a cluster on the bottom right. This structure motivates our approach, as the training weights contain information which may be useful at k-shot time, for instance given a few example from chestnut trees.

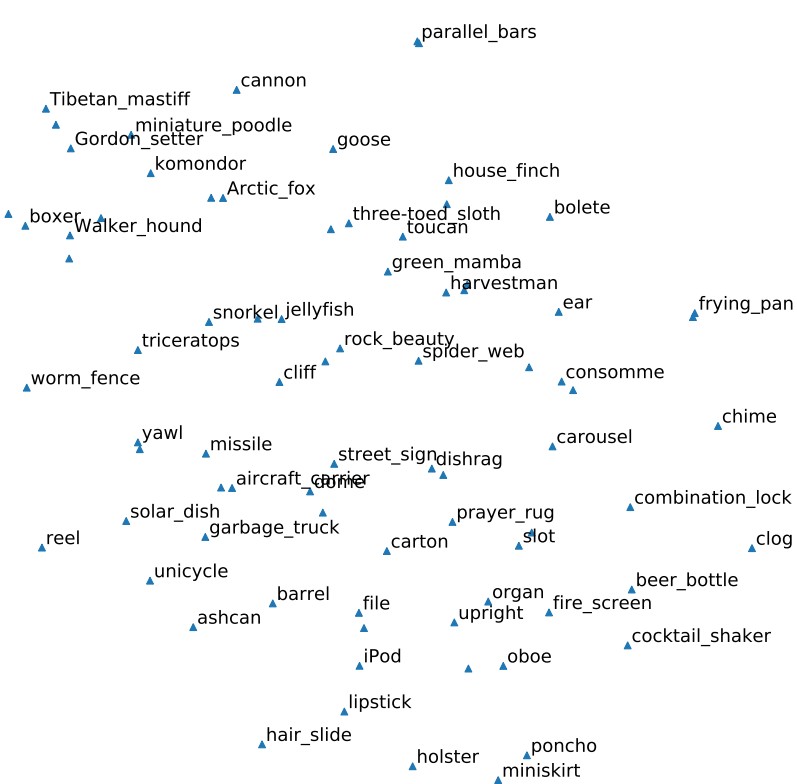

Figure 7: t-SNE embedding of the *mini*ImageNet weights trained using a ResNet-34 architecture. Structure is still present and we observe meaningful patterns, even though the classes in *mini*ImageNet are more unique than in CIFAR-100. For instance, goose, house finch, toucan, Arctic fox, green mamba and other animals are clustered on the top, with birds close to each other. Examples of other small clusters include poncho and miniskirt, or organ and oboe. For readability, not all class names are plotted.

# E  EXTENDED EXPERIMENTS

## E.1  T-SNE EMBEDDING OF THE WEIGHTS

We provide t-SNE embeddings for the weights of a VGG network trained in CIFAR-100 and a ResNet-34 trained on *mini*ImageNet. A structure in the weights is apparent and provides motivation for our framework. The results can be seen in Fig. 6 and Fig. 7.

## E.2  EXTENDED RESULTS ON *mini*IMAGENET

Fig. 8 provides extended results on k-shot learning for the *mini*ImageNet dataset for different network architectures. We investigate the influence of different feature extractors of increasing complexity and training data size on performance on: i) a VGG style network trained on 500 images per class, ii) a ResNet-34 trained on 500 examples per class, and iii) a ResNet-34 trained on all 600 examples per class.

## E.3  CHOICE OF REGULARISATION CONSTANT

Fig. 9 reports accuracy and calibration in terms of Expected Calibration Error (ECE) (lower is better) and log likelihoods (higher is better) for different regularisations of logistic regression for all three model architectures considered.

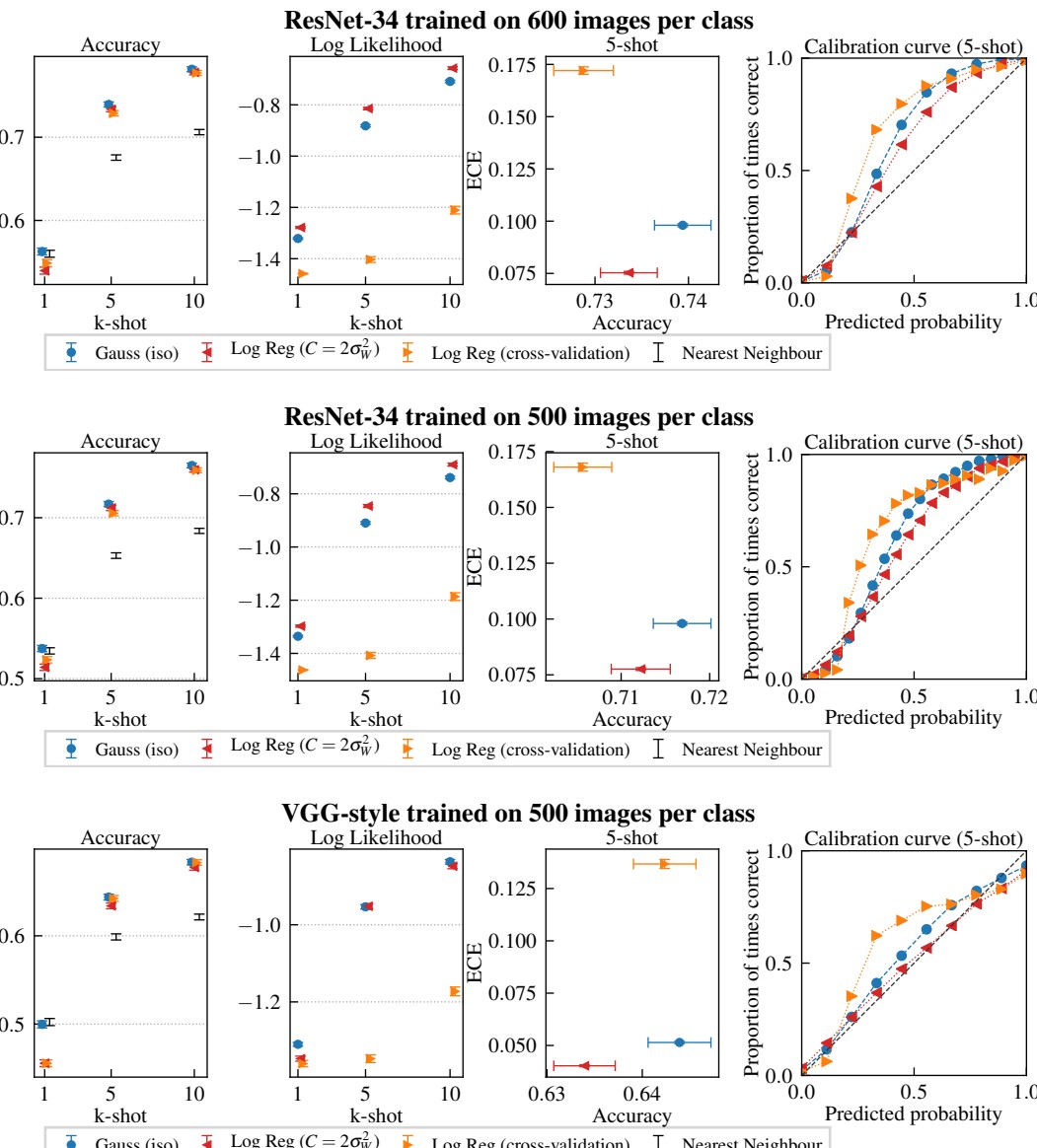

Figure 8: Extended results for the *mini*ImageNet dataset utilising different network architectures and representational training. *top:* a ResNet-34 trained with all 600 examples per class; *middle:* a ResNet-34 trained with 500 images per class; *bottom:* a VGG style network trained with 500 images per class. We highlight that for all three architectures the order of the different methods as well as the main messages are the same. However, the general performance in terms of accuracy and calibration differ between the architectures. The more complex architecture trained on most images performs best in terms of accuracy, indicating that it learns better features for k-shot learning. Both ResNets behave very similarly on calibration whereas the VGG-style network performs better (lower ECE and higher log likelihood as well as more diagonal calibration curve). This is in line with observations by Guo et al. (2017) that calibration of deep architectures gets worse as depth and complexity increase.

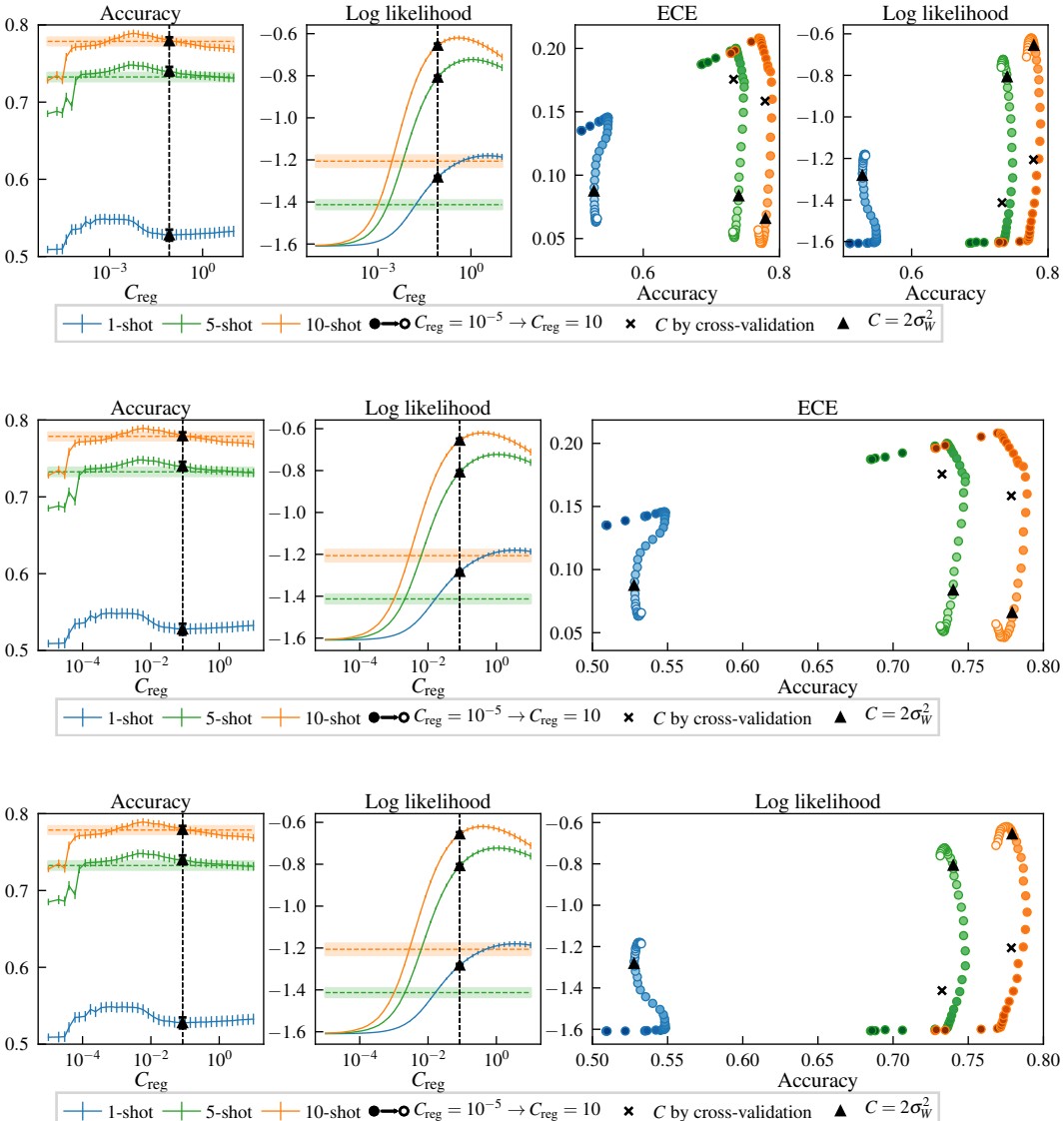

Figure 9: Choice of regularisation constant for logistic regression on k-shot learning. Note that all three rows use the same raw data that are only visualised differently. *Top:* Summary of accuracy and calibration in terms of log likelihood and Expected Calibration Error (ECE). *Middle:* detailed plot of ECE vs. accuracy. *Bottom:* detailed plot of log likelihood vs. accuracy. Results for $C_{\text{reg}} = 2\sigma^2_{\widetilde{W}}$ are drawn as black triangles. Dashed lines correspond to logistic regression with cross-validated (changing) regularisation constant. Colour brightness of the markers ranges from dark ($C = 10^{-5}$) to bright ($C = 10$). In addition to Fig. 4 we also provide results for calibration in terms of ECE (lower is better), which are consistent with log likelihoods (higher is better): The Bayesian inspired choice of the regularisation parameter strikes a good balance between accuracy and calibration and consistently outperforms cross-validated choice of the parameter.

### E.4 MODEL ASSESSMENT IN CIFAR-100

This section reports an extensive model comparison on CIFAR-100, both for the model of the weights $p(\mathrm{W} \,|\, \widetilde{\mathrm{W}})$ and for the inference procedure at k-shot time (MAP or Hybrid Monte Carlo (HMC) sampling using NUTS (Hoffman & Gelman, 2014), see the description of approximate inference algorithms in Appendix B). We report log-likelihood of the weights under different models, as well as accuracy, log-likelihood and calibration in a k-shot learning task. Tab. 7 and Tab. 8 show descriptions of the methods analysed for respectively phase 2 (concept learning) and phase 3 (k-shot learning) of our k-shot pipeline described in Sec. 2.1.

| Method name | Phase 2: Concept learning | |
| --- | --- | --- |
| | Prior distribution | Inference |
| Gauss (iso) | Gaussian isotropic covariance | MAP |
| Gauss (MAP prior) | Gaussian isotropic covariance | MAP |
| Gauss (integr. prior) | Gaussian full covariance | Integrated |
| GMM (supercl.) | GMM on superclasses iso. cov. | MAP |
| GMM (3, iso) | GMM on 3 clusters iso. cov. | MLE |
| GMM (3, diag) | GMM on 3 clusters diagonal cov. | MLE |
| GMM (10, iso) | GMM on 10 clusters iso. cov. | MLE |
| Laplace (diag) | Laplace diagonal covariance | MLE |

Table 7: Description of the inference for the parameters of the prior in phase 2 (concept learning) for the models in from Fig. 10. This specifies the inference procedure for $\theta$ in $p(\mathbf{w} \,|\, \theta)$ after observing the training weights $\widetilde{\mathrm{W}}$.

| Method name | Phase 3: k-shot learning | |
| --- | --- | --- |
| | Prior distribution | Inference |
| Gauss (iso) MAP | Gaussian | MAP |
| Gauss (MAP prior) MAP | Gaussian | MAP |
| Gauss (MAP prior) HMC | Gaussian | HMC |
| Gauss (integr. prior) MAP | Gaussian | MAP |
| Gauss (integr. prior) HMC | Gaussian | HMC |
| GMM (supercl.) MAP | GMM on superclasses | MAP |
| GMM (3, iso) MAP | GMM on 3 isotropic comp. | MAP |
| Laplace (diag) HMC | Laplace (diagonal) | HMC |
| Laplace (diag) MAP | Laplace (diagonal) | MAP |

Table 8: Methods and inference procedure during phase 3 (k-shot learning) for the models used in Fig. 10. This specifies the inference procedure used when computing $p(\mathrm{W} \,|\, \mathcal{D}, \widetilde{\mathrm{W}})$ for the specified prior distribution.

In the main text, we only consider an isotropic Gaussian model with MAP inference since we do not observe benefits from using alternative methods in terms of k-shot performance and calibration. Moreover, while we report results on a VGG-like architecture, we could also use a ResNet architecture, and preliminary results point to the same conclusion as experiments on *mini*ImageNet when switching from VGG to ResNet: the deeper features consistently lead to higher k-shot performance on all methods whereas the ordering of the methods stays roughly the same.

**Analysis of the models on held-out training weights.** First, we analyse how well the different prior models for the new softmax weights are able to fit the $\widetilde{C}$ training weights $\widetilde{\mathrm{W}}$. We randomly excluded 10 of those weights and evaluated their held-out negative log likelihood given the remaining $C - 10$ weights. We emphasise that this approach also constitutes a principled way to set the hyperparameters of the prior and, critically, relies on an explicit probabilistic model.

The negative log probabilities are averaged over 50 random splits and results of best optimised values w.r.t. hyperparameters are shown in Tab. 9 for CIFAR-100 (lower is better). We find that all models

| Model | Optimised value of mean negative log probability | |
|---|:---:|---|
| Gauss (iso) | | $-175.9 \pm 0.3$ |
| Gauss (MAP prior) | | $-196.1 \pm 0.5$ |
| Gauss (integr. prior) | | $-200.6 \pm 0.4$ |
| GMM 3-means (iso) | | $-179.0 \pm 0.3$ |
| GMM 3-means (diag) | | $-181.2 \pm 0.3$ |
| GMM 10-means iso | | $-181.6 \pm 0.4$ |
| GMM 10-means (diag) | | $-181.6 \pm 0.4$ |
| Laplace (iso) | | $-173.8 \pm 0.4$ |
| Laplace (diag) | | $-176.6 \pm 0.5$ |

Table 9: Held-out log probabilities on random 70/10-splits of the training weights for the different models on CIFAR-100. Values are averaged over 50 splits.

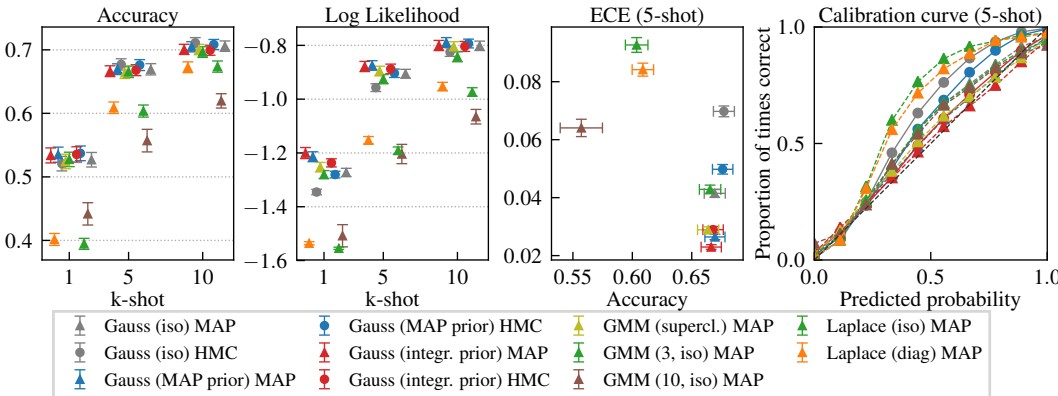

Figure 10: Results on CIFAR-100 for VGG style architecture. We report accuracy, log-likelihood and calibration for the methods and inference procedures presented in Tab. 8. With the exception of GMM (10, iso) and Laplace, all methods are similar terms of accuracy and log-likelihood. Gauss (integr. prior) HMC and Gauss (MAP) HMC are slightly better calibrated than our proposed Gauss (MAP) iso, but require significantly more computation for the sampling procedure.

behave very similar but that multivariate Gaussian models generally outperform other models. We attribute the good performance of the simpler models to the small number of data points ($C - 10 = 70$ training weights) and the high dimensionality of the space, which entail that fitting even simple models is difficult. Thus, more complicated models cannot improve over them.

**k-shot performance in CIFAR-100.** Accuracies are measured on a 5-way classification task on the k-shot classes for $k \in \{1, 5, 10\}$. Results were averaged two-fold: (i) 20 random splits of the 5 k-shot classes; (ii) 10 repetitions of each split with different k-shot training examples. Among our models, no statistically significant difference in accuracy is observed, with the exception of Laplace MAP and GMM (iso), which consistently underperforms. These findings are consistent in terms of log-likelihoods, see the first and second plots in Fig. 10.

Finally, our methods are generally well calibrated, with Gaussian models generally better than Laplace models. Moreover, all methods (with the exception of Laplace and GMM (10, iso) have low ECE and high accuracy, see the third and fourth plots of Fig. 10. While Gauss (integr. prior) HMC and Gauss (MAP) HMC are sightly better calibrated than our proposed method in the main paper, Gauss (MAP) iso, we believe the gain in calibration is not worth the significant increase in computational resources needed for the sampling procedure. Interestingly, both GMM approaches are not able to outperform the other, simpler models. This is in line with the previous observation that the simpler models are better able to explain the weights. Again, we attribute this inability of mixture models to use their larger expressivity/capacity to the small number of data points and the high-dimensionality

of weight-space which means learning even simple models is difficult. These observations suggest that the use of mixture models in this type of k-shot learning framework is not beneficial and is in contrast to the approach of Srivastava & Salakhutdinov (2013), who employ a tree-structured mixture model. The authors show compare a model in which the assignments to the superclasses in the tree are optimized over against a model with a naive initialisation of the superclass assignments, and show that the first outperforms the second. However, they do not compare against a simpler baseline, e.g., a single Gaussian model.

Overall, we observe that there is no significant benefit of more complex methods over the simple isotropic Gaussian, either in terms of accuracy, log-likelihood or calibration. Thus, our recommendation is that practitioners should use simple models and employ simple inference schemes to estimate all free parameters thereby avoiding expending valuable data on validation sets

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
