# OpenReview forum: "Discriminative k-shot learning using probabilistic models"
_ICLR.cc/2018/Conference — Reject_

### Official Review · AnonReviewer1 · 2017-11-25
**interesting but perhaps incremental**

**Rating:** 5
**Confidence:** 3

**Review:**

This paper presents a procedure to efficiently do K-shot learning in a classification setting by creating informative priors from information learned from a large, fully labeled dataset.  Image features are learned using a standard convolutional neural network---the last layer form image features, while the last set of weights are taken to be image "concepts".  The method treats these weights as data, and uses these data to construct an informative prior over weights for new features.

- Sentence two: would be nice to include a citation from developmental psychology.

- Probabilistic modeling section: treating the trained weights like "data" is a good way to convey intuition about your method.  It might be good to clarify some specifics earlier on in the "Probabilistic Modeling" paragraph, e.g. how many "observations" are associated with this matrix.

- In the second phase, concept transfer, is the only information from the supervised weights the mean and estimated covariance?  For instance, if there are 80 classes and 256 features from the supervised phase, the weight "data" model is 80 conditionally IID vectors of length 256 ~ Normal(\mu, \Sigma).  The posterior MAP for \mu and \Sigma are then used as a prior for weights in the K-shot task.  How many parameters are estimated for

  * gauss iso: mu = 256-length vector, \sigma = scalar variance value of weights
  * log reg: mu = 256-length zero vector, \sigma = scalar variance value of weights
  * log reg cross val: mu = 256-length zero vector, \sigma = cross validated value

If the above is correct, the information boosts K-shot accuracy is completely contained in the 256-length posterior mean vector and the scalar weight variance value?

- Is any uncertainty about \mu_MAP or \Sigma_MAP propagated through to uncertainty in the K-shot weights?  If not, would this influence the choice of covariance structure for \Sigma_MAP? How sensitive are inferences to the choice of Normal inverse-Wishart hyper parameters?

- What do you believe is the source of the mis-calibration in the "predictied probability vs. proportion of times correct" plot in Figure 2?

Technical: The method appears to be technically correct.

Clarity: The paper is pretty clearly written, however some specific details of the method are difficult to understand.

Novel: I am not familiar with K-shot learning tasks to assess the novelty of this approach.

Impact: While the reported results seem impressive and encouraging, I believe this a relatively incremental approach.

---

> ### Author Response · Authors · 2017-12-19
> **Reply to AnonReviewer1**
>
> Thank you very much for your review; we reply to it below. Please also read the general reply above.
>
> ad probabilistic model section) Thank you for the comment, we will incorporate this in the text.
>
> ad transfer and dimension of parameters) This is correct, we will make it clearer in the text.
>
> ad uncertainty of parameters) No, in our case \mu_MAP and \Sigma_MAP correspond to point estimates, which do not carry uncertainty; however, our framework also allows for other inference methods, such as MCMC sampling or variational inference, which do not require point estimates and use samples from the entire distribution.
>
> ad miscalibration) The problem of the miscalibration of neural networks is well known and has, for example, been analysed in [Guo et al. 2017]. We are not immune to this shortcoming of deep classifiers; however, we show that we are calibrated better than other methods.

---

### Official Review · AnonReviewer2 · 2017-11-27
**interesting interpretation of regularized Logistic regression but missing details and a very strong assumption**

**Rating:** 5
**Confidence:** 3

**Review:**

Authors present a k-shot learning method that is based on generating representations with a pre-trained network and learning a regularized logistic regression using the available data.  The regularised regression is formulated as a MAP estimation problem with the prior estimated from the weights of the original network connected final hidden layer to the logits — before the soft-max layer.

The motivation of the article regarding “concepts” is interesting.  It seems especially justified when the training set that is used to train the original network has similar objects as the smaller set that is used for k-shot learning.  Maps shown in Figures 6 and 7 provide good motivation for this approach.

Despite the strong motivation, the article raises some concerns regarding the method.
1. The assumption about independence of w vectors across classes is a very strong one and as far as I can see, it does not have a sound justification.  The original networks are trained to distinguish between classes.  The weight vectors are estimated with this goal. Therefore, it is very likely that vectors of different classes are highly correlated. Going beyond this assumption also seems difficult.  The proposed model estimates $\theta^{MAP}$ using only one W matrix, the one that is estimated by training the original network in the most usual way.  In this case, the prior over $\theta$ would have a large influence on the MAP estimate and setting it properly becomes important.  As far as I can see, there is no good recipe presented in the article for setting this prior.
2. How is the prior model defined?  It is the most important component of the method while precise details are not provided.  How are the hyperparameters set?  Furthermore, this detail needs to be in the main text.
3. With the isotropic assumption on the covariance matrix, the main difference between logistic regression, which is regularized by L2 norm and coefficient set proportional to the empirical variance, and the proposed method seems to be the mean vector $\mu^{MAP}$.  From the details provided in the appendix — which should be in the main text in my opinion — I believe this vector is a combination of the prior and mean of w_c across classes.  If the prior is set to 0, how different is this vector from 0?  Authors should focus on this in my opinion to explain why methods work differently in 1-shot learning.  In the other problems, the results suggest they are pretty much the same.
4. Authors’ motivation about concepts is interesting however, if the model bases its prediction on mean of w_c vectors over classes, then I am not sure if authors really achieve what they motivate for.
5. Results are not very convincing.  If the method was substantially different than baseline, I believe this would have been no problem.  Given the proximity of the proposed method to the baseline with regularised logistic regression, lack of empirical advantage is an issue.  If the proposed model works better in the 1-shot scenario, then authors should delve into it to explain the advantage.

Minor comments:
Evaluation in an online setting section is unclear.  It needs to be rewritten in my opinion.

---

> ### Author Response · Authors · 2017-12-19
> **Reply to AnonReviewer2**
>
> Thank you very much for your review; we reply to it below. Please also read the general reply above.
>
> ad 1) $\theta^{MAP}$ is estimated using the W matrix from training, but this matrix contains C samples from p(W|\theta). The importance of the prior vanishes as C increases and is not so fundamental.
> The softmax likelihood does have an identifiability problem: if all weights are shifted by the same offset then the same probabilities will result. By itself, this can result in dependencies in the weight’s posterior. However, the L2 regularisation applied in the first phase of learning (representational learning) mitigates this effect. Moreover, these dependencies have no effect on the quality of prediction, since by definition predictions are the same for these settings of the weights. We do not believe there are strong additional dependencies between the top-level weights, once the lower layers are fixed. Indeed, once the lower layers are fixed, the average hidden layer activation for a class indicates what a ‘good’ setting of the softmax weight for that class will be: the weight should simply lie in the direction of this vector (see [10] for a similar argument). A related observation lends some support to this argument: pilot experiments showed that -- when the lower level of the network is fixed and the top level weights are retrained several times to classify a class in the context of sets of different randomly selected classes -- the same weight vector is recovered each time for the common class. The conclusion is that the context does not matter, but rather just the representation of that class at the hidden layer. This does not speak directly to dependencies between the weight vectors of different classes, but it is consistent with this hypothesis and may explain the very strong performance of this seemingly overly-simple approach.
>
> 2) We use a Normal-Inverse-Wishart prior, which is the standard conjugate prior to a Gaussian model and has four hyperparameters, mu_0, kappa_0, Lambda_0, and nu_0. Standard approaches to set these hyperparameters are discussed in [Murphy 2012]; we try two approaches: 1) a weakly data dependent prior, 2) a prior that is set by cross-validation of log probabilities on the weights (see discussion in the supplement). Both approaches yield similar results, and especially for the isotropic model, the results are not very sensitive to the choice of prior parameters. We will clarify in the final version.
>
> 3) This is correct; if mu_0is zero, the size of kappa_0 determines how different mu_MAP will be to zero. Typically, this value is chosen to be (much) smaller than one (Murphy 2012), such that mu_MAP is non-zero, even in the one-shot case.
>
> 4) We agree that the concept transfer is limited to very few parameters in our experiments. Our experiments on CIFAR (e.g., Figure 10) show that there is no advantage, on this dataset, for using a more structured model, such as a mixture of Gaussians. However, the framework we present is general and allows for more elaborate probabilistic models leading to more ambitious concept transfer. Models such as the Gaussian latent feature model presented in Section 5.1 of [Griffiths, Ghahramani 2011] could be considered. Our results are a first step in this direction, and we see this as an exciting direction of future work for datasets with a higher number of classes.
>
> 5) We achieve state-of-the-art results by a large margin over competing methods, and the success of our approach should be measured against the current reference methods in the literature. Previous state of the art papers only compare to nearest neighbours with a very shallow network and, in particular, never compare to logistic regression, which is a much stronger baseline. Our approach is orthogonal to previous methods. On the considered datasets, learning deep features already helps the baselines (NN and LRCV) to beat other state of the art methods. We aimed to convey that our choice of probabilistic model is closely related to logistic regression, which we do not deem a disadvantage. Indeed, one message of the paper is that something as simple as logistic regression can beat all current methods, which has important implications for k-shot learning problems and how they should be tackled. We do not increase in accuracy substantially, but we get better calibration, which is often desirable (e.g., situations where making mistakes is expensive, such as self driving cars). Moreover, the results clearly show that using standard cross-validation leads to worse accuracy and much worse calibration, and, in particular, is not possible for 1-shot learning. In this regard, logistic regression using the weight variance as regularization is a proposed method, and not a baseline, which we will stress in the main text.
>
> 6) Thank you for pointing out that this section is not clear enough; we are happy to improve it. Could you please elaborate briefly on the aspects that are unclear?

---

### Official Review · AnonReviewer3 · 2017-12-04
**A simple probabilistic method to transfer knowledge from large datasets to smaller datasets (k-shot per class).**

**Rating:** 5
**Confidence:** 3

**Review:**

The authors introduce a probabilistic k-shot learning model based on previous training of a CNN on a large dataset. The weights of the softmax layer of the CNN are then used as the MAP solution in a concept learning scenario to put a prior over the soft-max weights of the classifier (logistic regression) for the dataset with k-shot examples. The model is compared against other models for doing k-shot learning in miniImageNet, and against different versions of the same model for CIFAR-100.

The paper introduces a very simple idea that allows transfer knowledge from a network trained with a large-dataset to a network-trained with a smaller amount of data, the data with k-shot examples per class. This is not a general way to do k-shot learning, because it heavily depends on the availability of the large dataset where the weights of the soft-max function can be extracted. But it seems to work for natural image data.

How many data observations are necessary to estimate \widetilde{W}_{MAP} such that it is still possible to obtain superior performance in the k-shot learning problem? Did you try the methods of Table 1 for CIFAR-100? The experiments on this dataset use models that are variations of the same proposed model.

---

> ### Author Response · Authors · 2017-12-19
> **Reply to AnonReviewer3**
>
> Thank you very much for your review; we reply to it below. Please also read the general reply above.
>
> We agree that our approach requires a certain amount of old training classes (“a database”) to build good features, but so do competing methods, such as matching networks or prototypical networks. In such a case, our method is general: learn good features with this data, and build a simple probabilistic model on top of the learnt parameters. Good feature representations exist for many data modalities such as images or text. If not much data is available in the training classes, we are not aware of methods with any guarantees.
>
> We did not investigate the dependence on the number of training classes. Generally, more classes is always better as is the case for all competing methods.
> We did not run Matching Networks on CIFAR 100 as we used this dataset to compare different probabilistic models and not to compare against other methods.

---

### Author Response · Authors · 2017-12-19
**General reply to all reviewers**

We thank the reviewers for their insightful comments. We will address them point by point. However, it seems that we did not manage to convey the simple but important findings of our work and we would like to emphasise them again: The field of k-shot learning has received significant attention in the last years, and many benchmarks use image datasets such as miniimagenet, cifar100, or omniglot.  The most prominent methods so far are based on episodic training, which is believed to be necessary for performing well on these k-shot learning task. In our opinion, this leads to slow and overly complicated training procedures. Our work suggests that these complications are not necessary to tackle few shot learning, and that a simple baseline based on deep features generalises surprisingly well and beats episodic training approaches. Previous state-of-the-art papers only compare to nearest neighbours with a very shallow network and, in particular, never compare to logistic regression, which is a much stronger baseline. In our opinion, this observation together with careful analysis of different models can influence the direction of the field moving forward.
Some of the concerns are regarding the simplicity of the Gaussian model. We argue that this simple probabilistic model performs so well compared to more complex models due to the low number of training classes in the studied datasets. However, our method is more general, and applying more complex variants to datasets with a large number of classes is an exciting and promising direction for future research. In order to illustrate the performance of our method and to compare to other methods, we chose to consider miniImagenet, which has become a de-facto standard.

---

### Decision · Program_Chairs · 2018-01-29
**ICLR 2018 Conference Acceptance Decision**

**Decision:**

Reject

**Comment:**

This submission presents intriguingly good results on k-shot learning and I agree with the authors that the results are better than the presented previous work, and that the method is simple, so I took a deeper look into the paper despite the overall negative reviews. However, I think in its current form, the paper is not suitable for publication:

- The previous work, that the authors compare to, were not really using comparable architectures: in fact, likely much worse base models with fewer parameters etc. I think any future version of this work would need to control for architecture capacity, otherwise how can we be sure where the gains come from? To me, this is a major unknown in terms of the credit assignment for the great results.
- The authors should be comparing with MAML (and follow-up work) by Finn et al. (2017)
- I don't really understand why the authors claim to have no need for validation sets. That's a very strong claim: are ALL the hyper-parameters (model architectures etc) just chosen in another, principled way? This issue would definitely need to be addressed in a follow-up work.